# Effects of Obesity and Exercise on Hepatic and Pancreatic Lipid Content and Glucose Metabolism: PET Studies in Twins Discordant for BMI

**DOI:** 10.3390/biom14091070

**Published:** 2024-08-27

**Authors:** Martin S. Lietzén, Andrea Mari, Ronja Ojala, Jaakko Hentilä, Kalle Koskensalo, Riikka Lautamäki, Eliisa Löyttyniemi, Riitta Parkkola, Virva Saunavaara, Anna K. Kirjavainen, Johan Rajander, Tarja Malm, Leo Lahti, Juha O. Rinne, Kirsi H. Pietiläinen, Patricia Iozzo, Jarna C. Hannukainen

**Affiliations:** 1Turku PET Centre, University of Turku, 20521 Turku, Finlandjhannukainen@gmail.com (J.C.H.); 2Institute of Neuroscience, National Research Council (CNR), 35128 Padua, Italy; 3Department of Medical Physics, Turku University Hospital, 20520 Turku, Finland; 4Heart Centre, Turku University Hospital, 20520 Turku, Finland; 5Department of Biostatistics, University of Turku, 20520 Turku, Finland; 6Department of Radiology, Turku University Hospital and University of Turku, 20520 Turku, Finland; 7Turku PET Centre, Radiopharmaceutical Chemistry Laboratory, University of Turku, 20521 Turku, Finland; 8Turku PET Centre, Accelerator Laboratory, Åbo Akademi University, 20500 Turku, Finland; 9A.I. Virtanen Institute for Molecular Sciences, University of Eastern Finland, 70211 Kuopio, Finland; 10Department of Computing, University of Turku, 20521 Turku, Finland; 11Turku PET Centre, Turku University Hospital, 20520 Turku, Finland; 12Obesity Research Unit, Research Program for Clinical and Molecular Metabolism, Faculty of Medicine, University of Helsinki, 00014 Helsinki, Finland; 13Abdominal Center, Obesity Center, Endocrinology, University of Helsinki and Helsinki University Central Hospital, 00014 Helsinki, Finland; 14Institute of Clinical Physiology, National Research Council (CNR), 56124 Pisa, Italy

**Keywords:** ectopic fat, glucose uptake, insulin sensitivity, low-grade inflammation, pancreas

## Abstract

Obesity and sedentarism are associated with increased liver and pancreatic fat content (LFC and PFC, respectively) as well as impaired organ metabolism. Exercise training is known to decrease organ ectopic fat but its effects on organ metabolism are unclear. Genetic background affects susceptibility to obesity and the response to training. We studied the effects of regular exercise training on LFC, PFC, and metabolism in monozygotic twin pairs discordant for BMI. We recruited 12 BMI-discordant monozygotic twin pairs (age 40.4, SD 4.5 years; BMI 32.9, SD 7.6, 8 female pairs). Ten pairs completed six months of training intervention. We measured hepatic insulin-stimulated glucose uptake using [^18^F]FDG-PET and fat content using magnetic resonance spectroscopy before and after the intervention. At baseline LFC, PFC, gamma-glutamyl transferase (GT), and hepatic glucose uptake were significantly higher in the heavier twins compared to the leaner co-twins (*p* = 0.018, *p* = 0.02 and *p* = 0.01, respectively). Response to training in liver glucose uptake and GT differed between the twins (Time*group *p* = 0.04 and *p* = 0.004, respectively). Liver glucose uptake tended to decrease, and GT decreased only in the heavier twins (*p* = 0.032). In BMI-discordant twins, heavier twins showed higher LFC and PFC, which may underlie the observed increase in liver glucose uptake and GT. These alterations were mitigated by exercise. The small number of participants makes the results preliminary, and future research with a larger pool of participants is warranted.

## 1. Introduction

Obesity and a sedentary lifestyle are major public health concerns and both are risk factors for developing type 2 diabetes [1,2,3].

In addition, increased liver and pancreatic fat content (LFC and PFC, respectively), impairments in hepatic metabolism, and pancreatic beta-cell function are also associated with type 2 diabetes [4,5]. Moreover, liver fat accumulation and whole-body insulin resistance have been suggested to decrease liver glucose uptake [4,5,6]. Furthermore, PFC has been associated with whole-body insulin resistance [7,8] and impaired beta-cell function in some [9,10,11] (but not all) studies [12].

Exercise training has been shown to decrease LFC in both lean and obese individuals [13,14,15]. However, the effects of exercise training on liver insulin sensitivity seem to vary depending on the training modality; moderate-intensity continuous training is more effective than sprint interval training [16]. However, less research has been conducted on the effects of exercise training on PFC and beta-cell function. In a previous study, we have shown that both sprint interval training and moderate-intensity continuous training decrease PFC and improve beta-cell function in insulin-resistant individuals after short-term training. However, in this study, no correlation was found between changes in PFC and beta-cell function [12].

Genetic background plays an important role in an individual’s susceptibility to the risk of obesity and type 2 diabetes [17]. It also affects the response to exercise training at an individual level [18]. The health benefits of exercise training are often studied using epidemiological studies or cross-sectionally comparing sedentary subjects with elite athletes that easily give too “rosy a picture” and do not consider genetic factors. Monozygotic twins, which have the same inherited genes, investigate the pathophysiology of insulin resistance and the effects of exercise without the confounding effect of genetics. In addition, monozygotic twins often share the same environment and are exposed to the same conditions until adulthood. As such, they represent an effective model to evaluate the independent consequences of acquired obesity, and the metabolic response to a lifestyle intervention.

The level of obesity is associated with the presence of comorbidities [19,20]. The association between BMI and all-cause mortality is non-linear, with a nadir between 20 and 24 kg/mg^2^, after which, the association is steep and rises linearly with the increasing BMI [21,22]. Previous studies have estimated that each unit increase in BMI increases the incidence of type 2 diabetes by 11–33% [23,24]. The Netherlands Twin Register Biobank project identified 17 monozygotic twin pairs with a BMI discordance of ≥3 kg/m^2^ (follow-up 6 years) and found worse obesity-related blood biomarker profiles (glucose, insulin, total cholesterol, LDL, HDL, triglycerides, CRP, IL-6, sIL-6R, GGT) in heavier (compared to leaner) twins [25].

This study, independent of genetic factors in monozygotic twins discordant for BMI, aimed to examine the effects of obesity on insulin-stimulated liver glucose uptake, LFC, PFC, and pancreatic beta-cell function. We also studied whether obesity-induced metabolic impairments can be ameliorated by long-term regular exercise training. We used non-invasive imaging of 2-deoxy-2-[^18^F]fluoro-D-glucose ([^18^F]FDG) by positron emission tomography (PET), computed tomography, and magnetic resonance imaging (MRS). We hypothesized that, at baseline, the heavier twins would have a higher LFC and PFC, and a lower hepatic insulin-stimulated glucose uptake and beta-cell function [6]. We hypothesized that exercise training would produce a greater decrease in LFC and PFC, and an improvement in whole-body and hepatic insulin sensitivity, as well as beta-cell function, in the heavier (compared to the leaner) twins [12,16]. We also expected that the decline in ectopic fat would explain the functional improvements.

## 2. Materials and Methods

### 2.1. Study Design

This study is an exercise training intervention that was conducted at the Turku PET Centre (Turku, Finland) as part of a larger study: Systemic cross-talk between brain, gut, and peripheral tissues in glucose homeostasis: the effects of exercise training (CROSSYS) (NCT03730610) [26]. The confounding effects of genetics were minimized by studying monozygotic twin pairs who had discordant BMIs. The clinical studies were carried out between 1/2019 and 10/2021 using high-quality imaging technology, such as PET and MRS. The clinical studies were discontinued when all the participants who met the inclusion criteria had completed the intervention. The study was conducted following Good Clinical Practices and the Declaration of Helsinki. All participants signed an informed consent before participating in the study. The study protocol, patient information, and informed consent for the CROSSYS intervention were approved by the Ethical Committee of the Hospital district of South-Western Finland (100/1801/2018/438§). Feasible requests for the datasets generated during the current study are available from the corresponding author. A detailed overview of the study design is depicted in Figure 1.

### 2.2. Participants

Monozygotic twins with discordant body weights were recruited from three population-based longitudinal twin studies as previously described [26]. All twins were identified through the Finnish central population registry. A total of 55 monozygotic pairs, discordant for BMI and/or insulin resistance, previously participated in the intensive metabolic study arm at the University of Helsinki, confirming their monozygosity by the genotyping of ten informative genetic markers [27]. Exclusion criteria were BMI > 60 kg/m^ 2^, body mass > 170 kg, waist circumference > 150 cm (due to the gantry limitations of PET and MRI machines), mental disorder or poor compliance, eating disorder or excess use of alcohol, active ulcer disease, diabetes requiring insulin treatment or fasting glucose > 10 mmol/L, pregnancy, a past dose of radiation, claustrophobia, presence of ferromagnetic objects that would make MRI contraindicated, physical disability that would prevent exercising, or any other condition that could potentially endanger the participant’s health during the study or interfere with the interpretation of the results. Twelve pairs met the inclusion criteria (monozygosity, within-pair difference of at least 2 in BMI, and at least one of the co-twins had a BMI over 25) [26] and ten pairs completed the study (Figure 2). Five of the leaner co-twins had impaired fasting glucose (IFG) and two had impaired glucose tolerance (IGT) (Table 1) as defined by the American Diabetes Association guidelines [28]. Seven of the heavier twins had impaired fasting glucose, two had impaired glucose tolerance, and two were treated for hypertension. Seven twin pairs were discordant for prediabetes status. None of the participants had diabetes, were treated for hyperlipidemia, or used diabetes medication. All the female participants were premenopausal. One of the heavier twins smoked and one used nicotine pouches. Two of the leaner twins smoked and two used nicotine pouches. None of the twins consumed more alcohol than the general safe limits (two units/day and three units/day for women and men, respectively). During the intervention, the participants were instructed to not alter their diet or lifestyle habits.

### 2.3. Training Intervention

All study participants performed the training intervention with a mean duration of 27 (SD 2) weeks; exercising was carried out four times a week at their place of residence, with one training session—chosen by the participant—supervised by a local personal trainer and the other three sessions consisting of unsupervised home-based exercise training, as detailed previously [26]. The training consisted of two endurance, one resistance, and one high-intensity interval training session per week. The day and time of the exercises were chosen by the participants to fit into their daily lives. The training intensity was increased progressively, and individual resistance training loads for ten-repetition sets were determined together with a personal trainer, corresponding to approximately 75% of the external load that could be lifted once, i.e., one-repetition maximum (1 RM). The resistance training loads were increased when sets of ten repetitions were easy to perform. The endurance training intensities were set as a percentage of the maximum heart rate measured during the VO_2peak_ test, and the training time and intensity were increased as the training intervention progressed, as previously shown [26].

The training program included two deload weeks in order to avoid overtraining and injuries. These deload periods were weeks 9 and 18. A heart rate monitor (Polar A370, Polar, Kempele, Finland) and a training log were used to monitor training adherence, intensity, and duration of individual workout sessions.

### 2.4. Blood Tests

Blood samples were collected before (Visit 1) and after the intervention (Visit 3) (Figure 1) from the antecubital vein. Fasting state (≥10 h) blood samples were collected between 8 and 10 am using VACUETTE or Vacutainer EDTA tubes, and plasma and serum samples were centrifuged according to the tube manufacturer’s recommendations. All blood samples were analyzed in the Turku University Hospital laboratory (Tykslab, Turku, Finland) on the same day except for FFA and HbA1c samples. FFA samples were frozen at (− 70 °C) and analyzed later in a larger batch. HbA1c samples were analyzed on the same day or during the next day at the latest. HbA1c samples were stored at +4 °C. Extra plasma and serum samples were taken to lithium heparin tubes and centrifuged according to the tube manufacturer’s recommendations. Extra samples were stored in a freezer (− 70 °C) for further analyses.

### 2.5. Euglycemic-Hyperinsulinemic Clamp and FDG-PET Study

The detailed FDG-PET-scan protocol was previously described [26]. In brief, liver-specific insulin-stimulated glucose uptake was studied by FDG-PET during an euglycemic-hyperinsulinemic clamp [29]. Participants fasted overnight (for at least 10 h) and avoided excess physical activity for 48 h before the FDG-PET study. They were placed in a supine position in the PET scanner (Discovery MI (DMI), GE Healthcare, Chicago, IL, USA) and instructed to avoid excess muscle contractions. The euglycemic-hyperinsulinemic clamp was performed, as originally described by Defronzo et al. [30]. Primed-constant insulin (Actrapid 100 U mL^− 1^, Novo Nordisk, Bagsværd, Denmark) infusion was started at a rate of 192 mU·m^− 2^·min^− 1^ during the first 4 min. Then, the infusion rate was reduced to 96 mU·m^− 2^·min^− 1^ for 4–7 min. After 7 min, the infusion rate was reduced to 48 mU·m^− 2^·min^− 1^ for the remaining clamp protocol. Once a steady state in the blood glucose was achieved, [^18^F]FDG was injected [26] and the liver PET scan started 43 (SD 1.6) minutes post-injection. Blood samples were taken at five-minute intervals to monitor blood glucose. During the euglycemic-hyperinsulinemic clamp, blood samples were also taken to evaluate insulin and free fatty acid levels. During the FDG-PET imaging, blood samples were also collected to determine plasma radioactivity (input function) [26]. Liver PET images were analyzed using Carimas software 2.10.3.0 (http://turkupetcentre.fi accessed on 23 September 2019). Activity values were averaged over three spherical volumes of interest (two in the upper and one in the lower half of the liver), as drawn in a total of 25 image planes). A fractional uptake rate was used to determine glucose uptake.

### 2.6. Magnetic Resonance Imaging and Magnetic Resonance Spectroscopy

Magnetic resonance imaging (MRI) and MRS were performed approximately four hours after lunch in order to define visceral fat mass, LFC, and PFC. The imaging was performed with a Siemens MAGNETOM Skyra fit 3 T MRI system (Siemens Healthcare, Erlangen, Germany), as described previously [26]. The magnetic resonance spectrum was collected from the target organ using a manually placed voxel. The LCModel program (version 6.3-1N) was used to differentiate the amplitudes of triglycerides in the frequency range of 0.9 to 2.8 ppm and the water in the MRS spectra. By dividing the triglyceride amplitudes by the sum of water and triglyceride amplitudes, LFC and PFC were estimated [31]. The visceral fat was segmented from the fat fraction maps by drawing 2-dimensional ROIs in every 5–9 slices and creating a 3-dimensional volume of interest from them using the interpolation feature of the Carimas software. Following this, all voxels with an intensity value below 0.5 (i.e., a fat fraction of over 50%) were excluded and the remaining volume of interest was considered as visceral adipose tissue.

### 2.7. Oral Glucose Tolerance Test

During visits 1 and 3 (Figure 1), a two-hour oral glucose tolerance test (OGTT) was performed after at least a 10 h fast. Blood samples were collected before, and at 15, 30, 45, 60, 90, and 120 min after ingestion of 250 mL of a solution containing 75 g of glucose (GlucosePro, Comed Oy, Ylöjärvi, Finland), to measure glucose, C-peptide, and insulin concentrations.

Beta-cell function parameters that included a basal insulin secretion rate, glucose sensitivity, rate sensitivity, and the 2 h to baseline potentiation factor ratio (PFR) were derived by using modeling previously described by Mari et al. [32,33]. The early- and late-phase insulin secretion rates (ISR early and ISR late) were calculated from the area under the curve from 0 to 30 min and from 30 to 120 min. Total ISR denotes the area under the curve of the whole 2 h OGTT.

### 2.8. Body Composition and Peak Aerobic Exercise Capacity Test

Body composition was measured using a bioimpedance analysis machine (Inbody 720; Biospace Co., Seoul, Republic of Korea). The machine measures participants’ total mass, skeletal muscle mass, body fat mass, body mass index, body fat percentage, and the lean mass balance between the left and right arm, left and right leg, and the trunk. Peak exercise capacity (V̇O_2peak_) was measured using a stationary bicycle ergometer test (Ergoline 800 s, VIASYS Healthcare, Hochberg, Germany). For men, the test starts with 50 W, which is increased by 30 W every 2 min until volitional exhaustion. For women, the test starts with 40 W, which is increased by 20 W every 2 min until volitional exhaustion. More in-depth details of the test have been previously described [26].

### 2.9. Statistical Analysis and Modeling

Normality assumption was checked from studentized residuals, and logarithmic transformations were performed to fulfill the normal distribution when needed, if possible. A linear mixed model for repeated time points using a compound symmetry covariance structure was used to analyze the data. The model included two within-factors: time (before and after intervention) and group (heavier and leaner co-twin), along with their interaction term (time*group). For differences between the groups at baseline, the same model was used but only the pre-intervention results were used. If after the intervention there was a significant time*group result, the same model was used to determine the within-group time effect to see if one of the groups had a significant time effect or if the groups responded differently to the intervention. Participants with missing data points were included in the statistical analysis by using the restricted maximum likelihood estimation within the linear mixed models. Hence, model-based means and 95% confidence intervals are reported. Correlation analyses were performed using Pearson’s product–moment correlation coefficient for normally distributed data and Spearman’s rank correlation coefficient for non-normally distributed data. All statistical tests were performed as two-sided and *p*-values less than 0.05 are considered statistically significant. *p*-values between 0.05 and 0.15 were interpreted as tendencies due to the small sample size. SAS system version 9.4 for Windows (SAS Institute, Cary, NC, USA) was used for the analyses.

## 3. Results

At baseline, the heavier twins had a lower M-value, V̇O_2peak_ (Table 2), and higher HbA1c (Table 3) compared to leaner co-twins. The exercise intervention improved VO_2peak_ and M-value in both groups (Time *p* < 0.05 in both variables). Exercise did not affect whole-body mass or fat percentage but tended to decrease visceral fat mass in both groups (Time *p* = 0.07).

LFC was significantly higher at baseline in the heavier twins compared to the leaner co-twins (*p* = 0.018) (Figure 3). The groups tended to respond differently to training (Time*group *p* = 0.09), in the heavier twins, the LFC tended to decrease with exercise (*p* = 0.11, while it was unchanged in the leaner twins (*p* = 0.69). At baseline, LFC correlated positively with alanine aminotransferase (ALT) and aspartate transaminase (AST) (*p* < 0.001 and *p* = 0.005, respectively (Table 4). In addition, the observed change, a tendency toward a lower LFC, correlated positively with the decrease in the GT (Figure 4 and Table 4). PFC was higher at baseline in the heavier twins compared to the leaner co-twins (*p* = 0.035) (Figure 3); however, after training, no changes were observed in the PFC of either group. In PFC, there was one individual (yellow) in the heavier group who saw an increase in PFC (+35.6%) against the general trend. The said participant’s visceral fat (−13.0%), fat mass (−7.6%), and LFC decreased (−52.4%), and VO_2peak_ (+1.5%) and the M-value (+75.3%) increased. In the leaner twins, one twin (dark green) went against the general trend with an increase in PFC (+845%). The said twin’s VO_2peak_ (+11%) and M-value (+82.4%) improved while visceral fat (+15.5%), fat mass (+37.5%), and LFC (+679.0%) increased.

Insulin-stimulated liver glucose uptake was higher at baseline in the heavier twins when compared to the leaner co-twins (*p* = 0.01) (Figure 5). The groups responded differently to training (Time*group *p* = 0.04), with a tendency toward a reduction in liver glucose uptake in heavier twins (*p* = 0.13), and no change in leaner twins (*p =* 0.56). In heavier twins, against the trend, glucose uptake increased in the two subjects (dark and light green). One (dark green) had an increase in VO_2peak_ (+11.4%) and M-value (+18.2%), a decrease in PFC (−33.3%), LFC (−22.3%), and visceral fat (−3.4%), but an increase in fat mass (+3.0%). The other one (light green) had an increase in VO2_peak_ (+4.9%) and M-value (+108.3%), a decrease in PFC (−37.4%) and LFC (−30.4%), but an increase in fat mass (+4.8%) and visceral fat (+4.2%).

Gamma-glutamyl transferase (GT) was higher at baseline in the heavier twins than the leaner co-twins (*p =* 0.02). The training response differed between groups (Time*group *p =* 0.004), with a significant GT reduction in the heavier twins (*p =* 0.032) and a tendency toward an increase in the leaner twins (*p =* 0.13) (Figure 4). GT values were within the normal range in both groups at baseline and after the intervention.

Basal and mean insulin were higher in the heavier twins compared to the leaner co-twins at baseline (*p =* 0.016 and *p =* 0.021, respectively). ISR early and total ISR were higher in the heavier twins compared to the leaner co-twins at baseline (*p =* 0.004 and *p =* 0.021, respectively) while ISR late only tended to be higher in the heavier twins compared with their leaner co-twins (*p =* 0.07) (Table 3).

At baseline, the M-value showed a strong negative correlation with basal and mean insulin and basal (t = 0 min), early (t = 0–30 min), late (t = 30–120 min), and total (t = 0–120 min) insulin secretion rate (*p* < 0.05 in all) (Table 4).

There was a statistically significant difference in training response between the twins in glucose sensitivity (Time*group *p =* 0.049) (Table 3), with the heavier twins showing no change (*p =* 0.36) and the leaner twins showing a decrease (*p =* 0.049) after the intervention.

## 4. Discussion

In this study, whole-body insulin resistance, LFC, and PFC were significantly higher in the heavier twins compared to the leaner co-twins. A six-month exercise intervention improved aerobic exercise capacity and whole-body insulin sensitivity but had no major impact on LFC or PFC. Interestingly, liver glucose uptake and GT were higher at baseline in the heavier twins. After the intervention, liver glucose uptake tended to decrease and GT decreased to the level of the leaner co-twins, post-training. Glucose sensitivity, a parameter representing the insulin response to different glucose concentrations, decreased after the intervention in leaner twins while no change was seen in heavier twins. Only 10 twin pairs completed the training intervention, which must be considered when interpreting the data.

Heavier twins had a higher LFC than their leaner co-twins at baseline. This is in line with previous research, showing that obesity is one of the main components leading to an increase in LFC [34]. The twin groups tended to respond differently to the training intervention, as the LFC tended to decrease (31%) only in the heavier twins. Previous studies have shown that exercise training decreases LFC, independent of weight loss [13,14]. However, there are also studies where the changes in LFC have been attributed to weight loss or a higher baseline LFC [16,35]. In our study, both groups at baseline had an average LFC above the upper normal limit of 5.56% [36] with the heavier twins having 8.2% higher LFC compared to the leaner twins. Post-training, only the visceral fat mass tended to decrease in both twin groups, and no change was observed in whole-body weight or fat percentage. This study shows that—independent of genetics—obesity leads to an excessive accumulation of lipids in the liver, and regular training is a drug-free method to reduce excess LFC.

Contrary to our hypothesis, the insulin-stimulated liver glucose uptake was significantly higher in the heavier twins compared with their leaner co-twins at baseline (Figure 5). Liver glucose uptake has been previously shown to be decreased in obesity [6] and type 2 diabetes [5] when compared to lean controls during euglycemic-hyperinsulinemia; this reflects the impaired glucokinase activity, i.e., the rate-limiting step in hepatic glycogen synthesis [37]. Under free-living (real-life) conditions, liver glucose uptake is elevated in leptin-receptor deficient pre-obese and obese rats compared to controls both in fasting and glucose-loaded states, strictly reflecting the degree of hepatic inflammation [38]. In this study, we documented that liver glucose uptake was an indicator of progression from steatosis to steatohepatitis. Liver glucose uptake is also increased in hepatic steatosis, when normalized to the metabolically active hepatic tissue, i.e., excluding the inert fat volume [39], and correlates positively with BMI [40]. 

Taken together, a higher liver glucose uptake in the heavier twins may reflect low-grade inflammation of the liver, as glucose uptake is typically increased in infiltrating inflammatory cells [39]. Exercise training may ameliorate this inflammation **[41]**. In fact, the heavier twins showed greater GT levels at baseline, and a significant reduction after exercise training (Figure 4), consistent with previous evidence [16]. GT is an oxidative stress marker, which is attributed to liver inflammation [42], correlating with body weight in this and other studies [43]. Together with the GT decline, we observed a dichotomic liver glucose uptake response in the twin pairs, with a liver glucose uptake decrease in the heavier twins, and a small increase in the leaner co-twins. In liver glucose uptake (Figure 5), two individuals in the heavier group saw changes in the opposite direction as the general trend (BMIs dark green 36.7 kg/m^2^ and light green 37.2 kg/m^2^). As they both had glucose uptake at the level of leaner twins, both improved their M-value and reduced their LFC and PFC. These changes may represent an overall improvement in insulin sensitivity, similar to that observed in the leaner group. 

In this study, GT correlated positively with LFC at baseline, and a significant positive correlation was observed between both the decrease in LFC and the decrease in GT, as observed in the heavier twins (Table 4). Overall, our findings suggest that exercise ameliorated the ectopic fat-induced liver inflammation in the heavier twins.

At baseline, PFC surpassed the upper normal limit of 6.2% [44] in both groups, although the heavier twins had a significantly higher PFC compared to the leaner twins (Figure 3). In this study, both groups were, on average, overweight due to the fact that the inclusion criteria were based on the BMI difference between twins within the pair, and not absolute BMI ranges. Our findings on PFC are in line with previous studies, showing that increasing obesity is progressively related to pancreatic fat accumulation [45].

Regular exercise training did not reduce PFC in either group. When assessed individually, one of the heavier twins was seen as yellow, and one of the leaner twins was seen as dark green, going against the average trend. The heavier twin had decreases in visceral fat, fat mass, and LFC. The leaner twin had increases in the same variables. While the increase in total fat mass may explain the increase in pancreatic fat in the leaner twin, in the heavier twins, the result may be due to unsuccessful voxel placement during the MRS assessment. In MRS, a voxel is placed into the target organ during the study while the subject is lying in the scanner. The MRS measurements are taken during breath holds to avoid organ movements; however, the small pancreas size and inaccurate voxel positioning might lead to detection errors, e.g., part of the voxel overlapping with intra-abdominal fat tissue. Also, pancreatic fat accumulation is usually not evenly distributed, which increases the importance of positioning the voxel in the same anatomical location before and after an intervention [46]. Thus, in this study, the challenges in voxel placement may explain the increase in PFC in the heavier twin as no increases in other fat parameters were observed. 

Studies on the effects of exercise training on PFC are sparse. In a previous study, we showed that a decrease in PFC can already occur after short-term training regardless of the baseline glucose tolerance [12]; while in a cross-sectional study in healthy monozygotic twins with discordant physical activity and fitness, no difference was observed between the groups in PFC [15]. The exact role of ectopic PFC on beta-cell function is also unclear, although pancreatic fat is more commonly increased in individuals with obesity and glucose tolerance impairment [8,9] than in lean normally-tolerant people. In rats, an increase in beta-cell triglyceride content has been shown to cause lipotoxicity and lipoapoptosis impairing beta-cell function, which may contribute to type 2 diabetes development [47]. However, the severity of organ steatosis induced in rodent models is usually not comparable to the levels occurring in humans.

At baseline, the heavier twins had significantly higher insulin secretion, as documented by the basal and mean insulin levels and the early and total insulin secretion rates. This was expected because obesity and insulin resistance induce compensatory hyperinsulinemia that might progress into type 2 diabetes [48]. Moreover, total ISR was strongly and inversely related to the M-value, suggesting that the greater insulin secretion observed in the heavier twins compared to the leaner co-twins is explained by insulin resistance. After the exercise intervention, there were no changes in any of the above-mentioned parameters. Eight months of vigorous exercise have been reported to reduce acute insulin secretion during an intravenous glucose tolerance test [49]. In this study, the M-value improvement was significant (Table 2), but training did not affect insulin secretion (Table 3), and the changes in M-value and insulin secretion were not significantly correlated to those in the total ISR. Consistent with insulin secretion data, glucose levels and HbA1c were unaffected by exercise. These data suggest that the sole improvement in insulin sensitivity in a six-month period is insufficient to establish a significant impact on glucose homeostasis in obese individuals without type 2 diabetes. A longer period of regular exercise or concomitant weight loss may be required.

These results, therefore, do not contrast with the established relationship between insulin secretion and insulin resistance, but the significant variability and the small group prevented precise assessment of the potential differences between the twin groups. However, our study shows that independent of genetics, obesity increases insulin secretion compared to leaner individuals, a phenomenon linked to the development of type 2 diabetes.

Glucose sensitivity derived from OGTT data reflects the slope of the dose response relating insulin secretion to glucose concentration. Namely, it indicates the amount of insulin produced by beta cells in response to different plasma glucose concentrations. At baseline, the twins did not differ in glucose sensitivity, consistent with previous studies [12]. After training, glucose sensitivity did not change in the heavier twins. In this study there were no participants with type 2 diabetes and the baseline glucose sensitivity was not low in either group.

Considering that the M-value was improved similarly in both groups, the above observations suggest that the M-value and glucose sensitivity are not necessarily correlated, as supported by previous observations [50]. The M-value is an estimate of peripheral glucose uptake measured using peripherally infused glucose during a typical hyperinsulinemic-euglycemic clamp study protocol. Glucose sensitivity, on the other hand, reflects the insulin response to blood glucose exposure, as in OGTT, glucose is absorbed from the intestine. Therefore, our results indicate that participants in our study have insulin resistance but no beta cell dysfunction. Overall, our study further strengthens the notion that independent of genetics, obesity increases pancreatic fat content and insulin secretion. However, the effect of pancreatic fat accumulation on pancreatic endocrine function needs further studies.

The limitations of this study include the small number of participants. Monozygotic twins discordant for BMI are uncommon, and a number declined to participate, partly due to the COVID-19 pandemic during the study period. Ideally, we would like to study monozygotic pairs with a substantial difference in body fat mass with leaner twins having a normal fat percentage. As we did not have measures of fat mass in the cohorts, we used BMI as the proxy to identify pairs with sufficient intrapair fat difference. In this study, we did not prescribe a homogeneous diet. Participants were instructed to complete a food diary and avoid changes in eating habits. Based on the diaries, no change in total caloric, fat, protein, or carbohydrate intake was observed post-training. However, as the diaries were filled by participants themselves, there is a possibility of underreporting, which could be a confounding factor. Ideally, nutrition would be carefully monitored by the researcher in real-time during the intervention. This is difficult, but in a shorter intervention, this could be possibly done. In this study, the intervention duration was six months and twin pairs were living all over the country. Of the ten twin pairs, eight were female. Ideally, there would be a balance between male and female participants. In order to confirm the preliminary results of the present study, future experiments would need to be conducted with a larger number of subjects, BMI discordance with a leaner group with normal fat percentage, balanced gender participation, and comprehensive diet planning. 

This study shows that the studied BMI-discordant twins differ significantly in several parameters and because of the minimized confounding effect of genetics, it indicates that lifestyle choices leading to obesity may have a significant effect on liver and beta-cell functions, independent of individual genetics. However, in the present study, the sample size was ten pairs, and in some individuals, the results were opposite to the general trend. Therefore, the current data should be regarded as preliminary, and future studies are warranted. LFC and insulin secretion during OGTT have been previously studied in monozygotic twin pairs discordant for BMI [51]; we have examined PFC in monozygotic twins discordant for physical activity [15], but to our knowledge, the novelty of the current study concerns the simultaneous evaluation of the effects of exercise on LFC, PFC, pancreatic, and liver metabolism in monozygotic twins discordant for BMI.

## 5. Conclusions

In BMI-discordant twins, a greater BMI was associated with increased liver and pancreatic fat content, liver glucose uptake, and the liver oxidative stress marker GT. Heavier twins responded to exercise training with a decrease in GT and a trend toward a decrease in liver glucose uptake, which likely reflects a reduction in liver inflammation. The findings underscore the impacts of phenotype and lifestyle factors in genetically identical individuals. Due to the small sample size, future studies with a larger number of subjects are warranted.

## Figures and Tables

**Figure 1 biomolecules-14-01070-f001:**
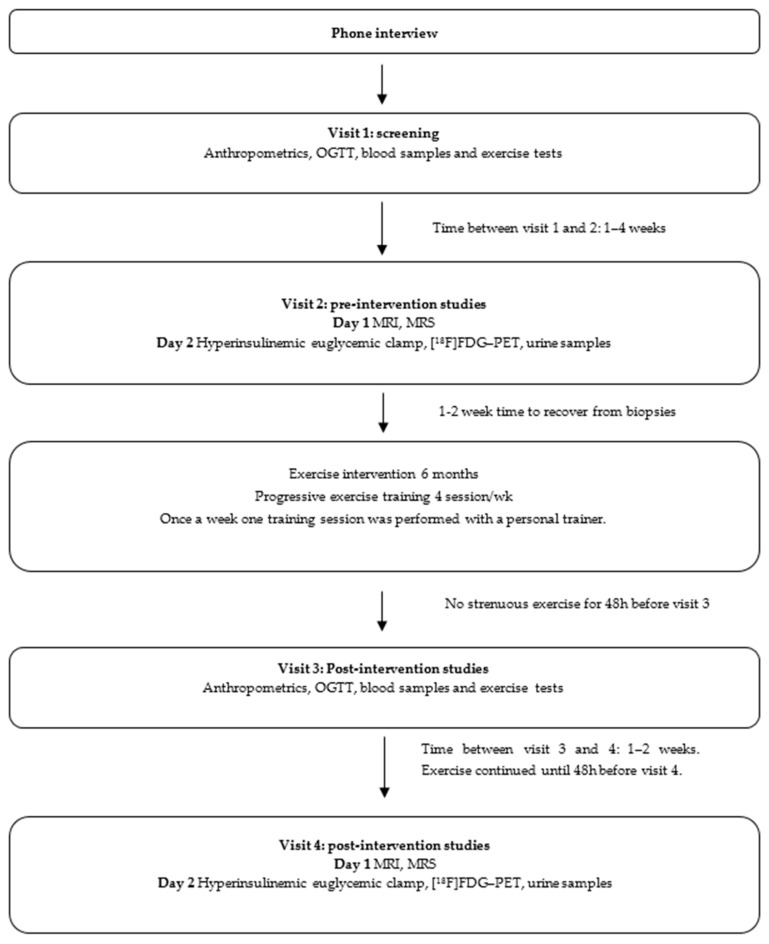
Overview of the study protocol. Abbreviations: [^18^F]FDG, 2–deoxy–2– [^18^F]fluoro–D–glucose; MRI, magnetic resonance imaging; MRS, magnetic resonance spectroscopy; OGTT, oral glucose tolerance test; PET, positron emission tomography.

**Figure 2 biomolecules-14-01070-f002:**
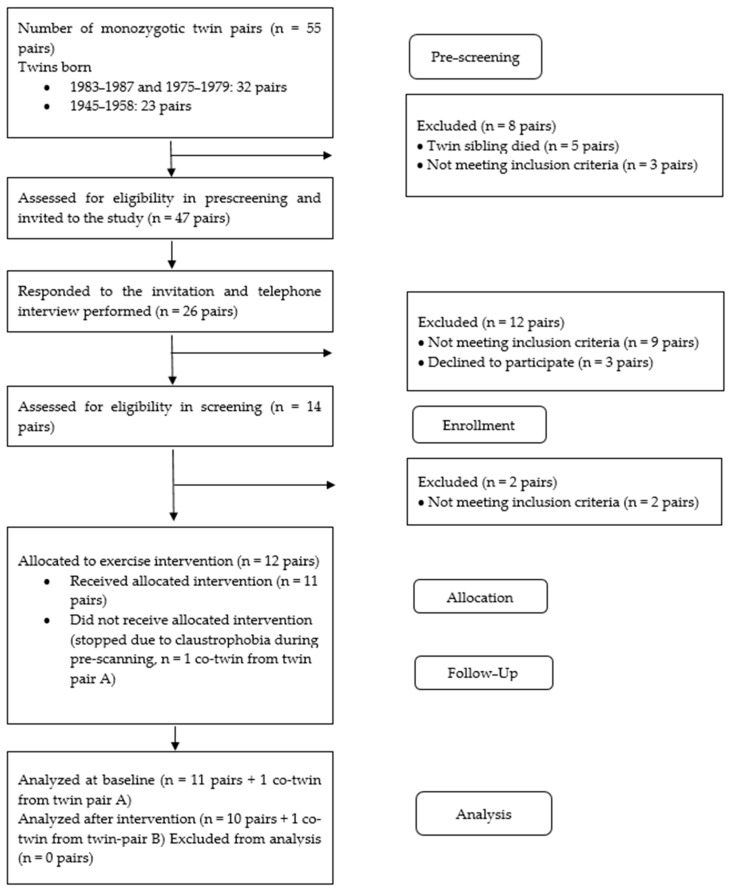
Consort flow.

**Figure 3 biomolecules-14-01070-f003:**
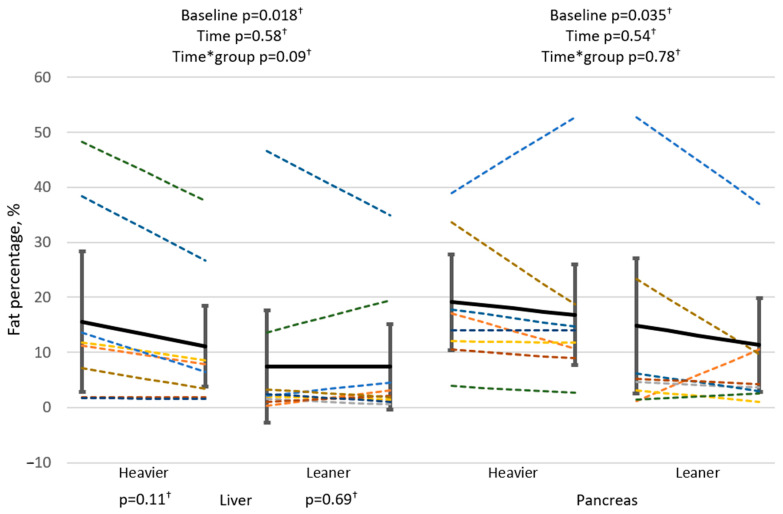
At baseline, the liver and pancreatic fat percentage was higher in the heavier twins compared to the leaner co-twins. In the heavier twins, liver fat percentage tended to decrease with exercise, while it was unchanged in the leaner twins. In the pancreatic fat percentage, no changes were observed after the intervention. Linear mixed model used for analysis. A single color represents a twin pair. Black line with confidence intervals represents the group average. Color coding is the same in both organs. Pancreatic fat percentage heavier twins: pre n = 8, post n = 9, and leaner twins: pre n = 10, post n = 9. Liver fat percentage heavier twins: pre n = 8, post n = 8, and leaner twins: pre n = 10, post n = 10, ^†^ Logarithmic transformation.

**Figure 4 biomolecules-14-01070-f004:**
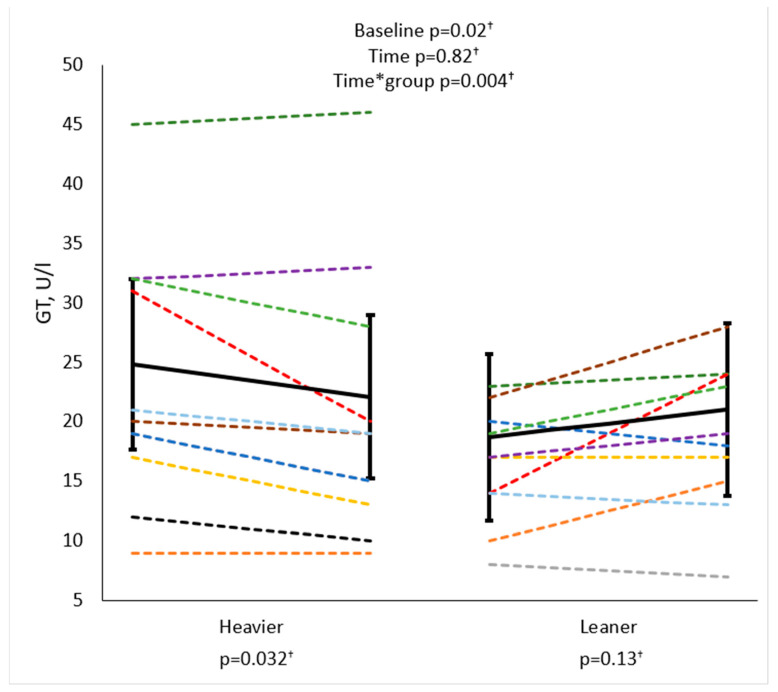
Gamma-glutamyl transferase (GT) was higher at baseline in the heavier twins and the effect of training differed between the twins; for the heavier twins, a decrease in GT was observed while an increase was seen for the leaner twins. Linear mixed model used for analysis. A single color represents a twin pair. Black line with confidence intervals represents the group average. Units per liter (U/l) Leaner twins: pre n = 12, post n = 11, and heavier twins: pre n = 12, post n = 10, ^†^ Logarithmic transformation.

**Figure 5 biomolecules-14-01070-f005:**
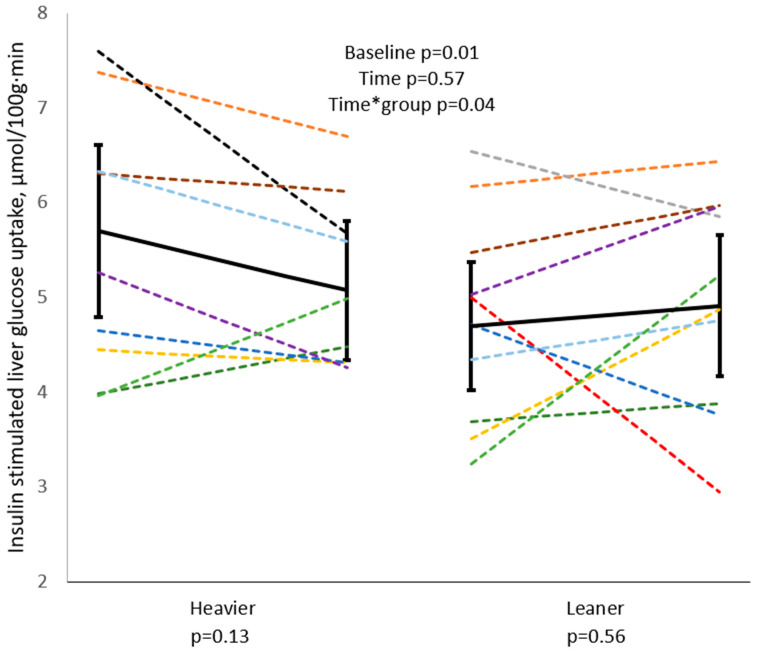
At baseline, liver glucose uptake was higher in the heavier twins compared to the leaner co-twins. The effect of exercise training differs between the twin groups, as training seems to decrease liver glucose uptake in the heavier twins while increasing it in the leaner co-twins. Linear mixed model used for analysis. A single color represents a twin pair. Black line with confidence intervals represents the group average. Heavier twins: pre n = 11, post n = 9, and leaner twins: pre n = 11, post n = 10.

**Table 1 biomolecules-14-01070-t001:** BMI of all participants and their glycemic status at baseline.

	Heavier		Leaner	
Twin Pair ID	BMI (kg/m^2^)	Glycemic Status	BMI (kg/m^2^)	Glycemic Status
1	42.6	normal	40.1	IFG
2	36.7	IFG	24.1	normal
3	30.8	IFG, IGT	25.4	normal
4	42.2	IFG	25.4	normal
5	33.8	IFG	27.8	IFG, IGT
6	30.1	normal	27.9	normal
7	47.0	IFG	42.7	normal
8	25.2	IFG, IGT	20.8	IFG, IGT
9	35.9	normal	27.4	IFG
10	47.7	IFG	29.3	IFG
11	31.1	normal	28.2	normal
12	37.2	normal	30.1	normal

IFG, impaired fasting glucose; IGT, impaired glucose tolerance.

**Table 2 biomolecules-14-01070-t002:** Subject characteristics (95% CI) of the leaner and heavier twin groups before and after exercise intervention.

	Heavier		Leaner		*p*-Value		
	Pre	Post	Pre	Post	Baseline	Time	Time*group
n	12	11	12	10			
Male/female	4/8	4/7	4/8	4/6			
Age, years	40.4 (37.5;43.4)		40.4 (37.5;43.4)				
Weight, kg	108.7 (91.8;125.7)	108.0 (93.1;122.9)	86.4 (72.4;100.3)	86.9 (72.6;101.2)	0.001 *	0.95	0.37
BMI, kg/m^2^	36.7 (32.2;41.1)	36.4 (32.4;40.4)	29.1 (25.1;33.1)	29.3 (25.3;33.2)	0.001 *	0.92	0.41
Waist circumference, cm	117.7 (106.3;129.2)	115.0 (106.8;123.1)	96.5 (84.7;108.3)	94.4 (82.3;106.6)	0.001 *	0.17	0.74
Fat-free mass, kg	35.9 (31.0;40.7)	36.2 (32.1;40.3)	33.1 (29.0;37.2)	33.9 (30.6;37.2)	0.003 *^†^	0.14	0.10
Fat mass, kg	45.5 (33.8;57.3)	44.5 (34.6;54.4)	27.8 (17.9;37.7)	26.9 (14.4;39.4)	0.001 *^†^	0.70 ^†^	0.97 ^†^
Visceral fat mass, kg	5.9 (4.5;7.3)	5.5 (4.4;6.5)	3.1 (2.0;4.3)	3.2 (2.0;4.4)	0.002 *^†^	0.07	0.29
Fat percentage, %	40.6 (35.5;45.7)	40.0 (35.9;44.1)	30.4 (24.0;36.9)	29.5 (20.3;38.7)	0.001 *	0.37	0.72
V̇O_2peak_, mL·kg^−1^·min^−1^	25.6 (22.7;28.5)	28.3 (26.1;30.6)	32.4 (27.3;37.4)	35.1 (29.9;40.2)	0.003 *	0.001 *	0.94
Triglycerides, mmol/L	1.4 (0.9;1.9)	1.2 (0.9;1.5)	0.8 (0.6;1.0)	0.8 (0.6;1.0)	0.040 *	0.54 ^†^	0.49 ^†^
FFA, mmol/L	0.59 (0.51;0.67)	0.68 (0.45; 0.91)	0.52 (0.34;0.69)	0.54 (0.14; 0.94)	0.29	0.63	0.63
ALT, u/L	33.8 (23.3;44.2)	31.4 (22.3;40.6)	31.6 (13.1;50.1)	31.5 (17.7;45.3)	0.30 ^†^	0.95 ^†^	0.42 ^†^
ALP, u/L	59.4 (51.4;67.4)	59.6 (51.5;67.8)	51.8 (43.2;60.3)	56.9 (47.2;66.7)	0.08 ^†^	0.31 ^†^	0.11 ^†^
AST, u/L	22.2 (18.7;25.6)	22.6 (18.3;26.9)	27.5 (15.8;39.2)	25.8 (14.8;36.9)	0.28 ^†^	0.86 ^†^	0.92 ^†^
CRP, mg/L	2.8 (1.4;4.2)	3.7 (1.3;6.2)	2.1 (0.3;3.9)	1.2 (−0.6;3.1)	0.050 *	0.69 ^†^	0.50 ^†^
M-value, μmol/kg*min	23.1 (16.3;30.0)	31.4 (20.4;42.3)	37.6 (26.7;48.5)	46.9 (31.7;62.1)	0.007 *	0.022 *	0.82

*p*-value (linear mixed model) for a baseline: within-pair difference before the intervention, time: pre- and post-difference in the whole group, Time*group: did the training response differ within twin pairs. For M-value heavier co-twins: pre n = 11, post n = 9, leaner co-twins: pre n = 11, post n = 10. Abbreviations: BMI, body mass index; VO_2peak_, aerobic capacity; FFA, free fatty acid; ALT, alanine transaminase; ALP, alkaline phosphatase; AST, aspartate aminotransferase; CRP, C-reactive protein; M-value, whole-body insulin sensitivity, * Statistically significant *p*-value (*p* ≤ 0.05). ^†^ Logarithmic transformation.

**Table 3 biomolecules-14-01070-t003:** OGTT and beta-cell function (95% CI) of the leaner and heavier twin groups before and after exercise intervention.

	Heavier		Leaner		*p*-Value		
	Pre	Post	Pre	Post	Baseline	Time	Time*group
n	12	11	12	10			
Basal glucose, mmol/L	5.7 (5.4;6.0)	5.8 (5.6;6.1)	5.5 (5.2;5.7)	5.5 (5.2;5.7)	0.39	0.37	0.42
Mean glucose, mmol/L	7.5 (6.7;8.2)	7.7 (7.0;8.3)	6.9 (6.1;7.8)	7.0 (6.1;7.9)	0.17	0.68	0.78
2 h glucose, mmol/L	6.5 (5.7;7.3)	6.6 (5.7;7.4)	6.1 (4.9;7.2)	5.8 (4.3;7.4)	0.32	0.84	0.52
Hba1c, mmol/mol	36.5 (35.1;37.9)	36.0 (34.2;37.7)	34.9 (32.8;37.0)	34.7 (32.2;37.1)	0.047 *^†^	0.58	0.68
Basal ins, pmol/L	70.5 (54.4;86.6)	63.5 (32.0;95.0)	44.0 (28.5;59.5)	49.0 (26.2;71.9)	0.006 *^†^	0.50 ^†^	0.71 ^†^
Mean insulin, pmol/L	371.6 (229.9;513.3)	339.4 (214.2;464.6)	235.8 (163.3;308.4)	258.2 (169.4;347.0)	0.01 *^†^	0.88 ^†^	0.66 ^†^
Basal insulin secretion rate, pmol∙min^−1^∙m^−2^	111.4 (97.5;125.2)	111.0 (95.4;126.5)	87.4 (69.4;105.4)	86.2 (69.6;102.7)	0.030 *	0.80	0.88
Glucose sensitivity, pmol∙min^−1^∙m^−2^∙mM^−1^	92.5 (69.4;115.6)	103.4 (75.3;131.5)	98.5 (77.5;119.5)	75.2 (48.6;101.9)	0.66	0.44	0.049 *
	*p* = 0.36		*p* = 0.049 *				
Rate sensitivity, pmol∙m^−2^ mM^−1^	1366.6 (906;1827)	1169.9 (680;1660)	1028.0 (668;1388)	1107.5 (601;1614)	0.049 *	0.70	0.13
PFR, dimensionless	1.4 (1.1;1.8)	1.3 (0.9;1.6)	1.7 (1.2;2.2)	1.5 (0.9;2.1)	0.48 ^†^	0.30 ^†^	0.52 ^†^
ISR early, nmol/m^2^	13.1 (9.7;16.6)	12.3 (8.8;15.8)	10.0 (7.1;12.9)	9.3 (6.3;12.3)	0.004 *	0.50	0.92
ISR late, nmol/m^2^	31.9 (28.5;35.4)	29.1 (25.7;32.6)	26.9 (20.9;32.9)	25.7 (20.2;31.3)	0.07	0.24	0.50
Total ISR, nmol/m^2^	45.1 (39.2;50.9)	41.5 (35.8;47.2)	36.9 (29.6;44.2)	35.1 (27.9;42.2)	0.021 *	0.23	0.52
2 h OGIS, mL∙min^−1^∙m^−2^	375.0 (354.1;395.8)	369.0 (349.8;388.2)	414.7 (384.3;445.1)	415.8 (389.9;441.6)	0.037 *	0.75	0.63

*p*-value (linear mixed model) for a baseline: within-pair difference before the intervention, time: pre- and post-difference in the whole group, Time*group: did the training response differ within twin pairs. Abbreviations: Hba1c, glycosylated hemoglobin; rate sensitivity, a parameter characterizing early insulin secretion; PFR, potentiation factor ratio; ISR early, insulin secretion rate at time 0–30 min; ISR late, insulin secretion rate at time 30–120 min; 2 h OGIS, oral glucose insulin sensitivity from 0–120 min. * Statistically significant *p*-value (*p* ≤ 0.05). ^†^ Logarithmic transformation.

**Table 4 biomolecules-14-01070-t004:** Correlations with liver fat percentage, liver glucose uptake, and insulin resistance.

	Baseline		Change	
	r	*p*	r	*p*
Liver fat percentage				
ALT, U/L	0.83	<0.001 *	0.49	0.05 *^‡^
AST, U/L	0.64	0.005 *		0.34
ALP, U/L	0.35	0.15	0.06	0.02 *
Liver glucose uptake				
ALT, U/L	−0.09	0.7 ^‡^	0.02	0.95 ^‡^
AST, U/L	−0.37	0.09 ^‡^	−0.06	0.79
ALP, U/L	−0.04	0.85	−0.18	0.45
GT, U/L	−0.40	0.06 ^†^	−0.02	0.92
GT				
Weight, kg	−0.09	0.68	0.46	0.04 *^‡^
BMI, kg/m^2^	0.03	0.90	0.47	0.03 *^‡^
Visceral fat, kg	0.54	0.021 *		0.12
Triglycerides, mmol/L	0.44	0.031 *^‡^	0.42	0.06
Glucose sensitivity				
Pancreatic fat content, %	−0.26	0.29	−0.08	0.77
M-value				
Basal insulin, pmol/L	−0.70	<0.001 *	−0.17	0.46
Mean insulin, pmol/L	−0.70	<0.001 *	−0.24	0.31
Basal insulin secretion rate, pmol∙min^−1^∙m^−2^	−0.70	<0.001 *	0.01	0.95
ISR early, nmol/m^2^	−0.61	0.002 *	−0.27	0.23
ISR late, nmol/m^2^	−0.76	<0.001 *	−0.05	0.84
Total ISR, nmol/m^2^	−0.83	<0.001 *	−0.15	0.52

Abbreviations: ALT, alanine transaminase; ALP, alkaline phosphatase; GT, gamma-glutamyl transferase; AST, aspartate aminotransferase; BMI, body mass index; basal insulin secretion rate; insulin secretion rate at time 0 min; ISR early, insulin secretion rate at time 0–30 min; ISR late, insulin secretion rate at time 30–120 min; total ISR, insulin secretion rate at time 0–120 min. * Statistically significant *p*-value (*p* ≤ 0.05) (Pearson’s product–moment correlation coefficient), ^†^ Logarithmic transformation, ^‡^ Spearman’s rank correlation coefficient.

## Data Availability

Feasible requests for the datasets generated during the current study are available from the corresponding author.

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
