# Peer review of "Effects of Obesity and Exercise on Hepatic and Pancreatic Lipid Content and Glucose Metabolism: PET Studies in Twins Discordant for BMI"

_biomolecules, 2024, doi:10.3390/biom14091070_

Round 1
Reviewer 1 Report (New Reviewer)
Comments and Suggestions for Authors
The manuscript is an original work investigating the impact of obesity and exercise on parameters related to visceral fat and glucose metabolism. Although the sample is limited due to the difficulty of meeting the inclusion criteria for each pair of twins, the preliminary results are interesting, showing the significant weight of lifestyles over genetics. However, improvements are required before publication:
Please, cite correctly the first sentence of the Introduction.
In the abstract, the authors should write “increased liver and pancreatic fat content (LFC and PFC, respectively)”, as in the introduction, since “increased liver (LFC)” is not appropriate.
The introduction is well structured. However, it is not clear what is in the existing literature concerning the key topic of this work (referring to the monozygotic twins), and the methodology does not clarify this.
Thus, the authors should better explain the interest in using monozygotic twins discordant for BMI and what magnitude of this difference should be considered. If possible, include supporting studies. In the 2.2. Participants subsection, the authors indicate “monozygosity, within-pair difference of at least 2 in BMI, and at least one of the pairs had a BMI over 24”, is that enough? Would the authors expect significant results within a BMI difference, for example, between 22.5 and 24.5? Moreover, why there is not a cut point for the BMI and BMI is not an inclusion criterion? It seems that the body weight was the inclusion criteria, is this correct? In the Discussion section, the authors indicate “the inclusion criteria was based on BMI difference between twins within the pair, and not absolute BMI ranges”. The description implies that pairs could be both normal weight and overweight/obese or overweight/obese and overweight/obese, why the authors did not choose monozygotic twins that were one normal weight and the other overweight/obese? Clarify all this. Additionally, since “five of the leaner co-twins had impaired fasting glucose and two had impaired glucose tolerance”, it is recommended to include a table with the BMI of each participant and the “glucose status”, or in table 2.
Moreover, the authors hypothesized that at baseline the heavier twins would have a higher LFC and PFC, and a lower hepatic insulin-stimulated glucose uptake and beta-cell function. Are there no previous studies achieving this? If so, indicate it—the same for the exercise hypothesis.
The list of Abbreviations should be continuous to the Fig. 1 title.
In Fig. 2, some information is missed: (stopped due to claustrophobia during…
In Training Intervention, please include briefly how the loads for resistance and the intensities for aerobic exercises were determined, and which parameter was used to establish the increases.
In the blood test, include the location of the punction (antebrachial vein?), if there was a specific hour (before 9 am?), and the used system (such as EDTA vacutainer). Was the whole blood stored at -70 ºC or plasma samples? If so, specify the centrifugation conditions.
If logarithmic transformations were performed to fulfill the normal distribution when needed, why Spearman’s rank correlation coefficient was used for non-normally distributed data? Was the normal distribution not achieved with the logarithmic transformations?
Check carefully for misspellings, such as “Table 1.” in lines 217, 218, 245, 247.
In Fig. 3 include the meaning of the colors of the lines, indicate what “a” means, and remove “†Logarithmic transformation.”, which is not necessary—the same for Fig. 4 and 5.
In lines 357-358, include the BMI for the 2 individuals in the heavier group who saw changes to the opposite direction as the general trend for the liver glucose uptake, since it is necessary to understand if the BMI was far or close to the leaners’ BMI.
Author Response
Thank you for your comments on the manuscript. Attached is the point-by-point reply to your comments/suggestions for improvement.

Reviewer 2 Report (New Reviewer)
Comments and Suggestions for Authors
I consider the topic being addressed an important one in the current context.
Materials and methods used are described accordingly.
I think that additional data is needed about the administered diabetes medication, especially if it could influence the results.
I also consider it necessary to present if there were changes in the diet during this period
It should be mentioned that in table 1 the ALT values ​​are entered twice.
The results and discussions are presented coherently.
Author Response
Thank you for your comments on the manuscript. Attached is the point-by-point reply to your comments/suggestions for improvement.

This manuscript is a resubmission of an earlier submission. The following is a list of the peer review reports and author responses from that submission.
Round 1
Reviewer 1 Report
Comments and Suggestions for Authors
The submitted paper aimed to understand whether obesity and regular exercise were associated with lipid content in the liver and pancreas, and with insulin-stimulated glucose uptake and beta-cell function. This study is part of a larger study with published papers and, as in the paper published in the journal Diabetes, Obesity and Metabolism in 2024 (DOI: 10.1111/dom.15311), a sub-sample of 10 twins who met the inclusion criteria and completed the study was analysed. In each pair, one of the twins was classified as heavier and the other as leaner, based on their BMI. The methodology considered for the statistical analysis is very similar to that considered in the published paper, as is the presentation of the results, but of course the outcomes are different. The paper is well written, but in my opinion the presentation and discussion of the analysis of the 10 twin pairs should be completed. The sample size of only 10 pairs of twins is very small to study tendencies, and this limitation should be taken into account when presenting and discussing the results, but at the same time it allows the researcher to make a deeper analysis of each pair, which was not considered.
Figures 3, 4 and 5 and corresponding text in the Results and Discussion sections: The figures should be more detailed to allow the reader to identify the heavier and corresponding leaner for each pair of twins. The analysis of each pair of twins shown in Figures 3, 4 and 5 should be presented in the Results section and discussed in the Discussion section. Only the general trends are presented and discussed, but it would be important to state that opposite trends were observed and to remember that these trends were obtained with a very small sample. The figures show that some individuals have very different patterns to the others. Who are these individuals? Do they have different characteristics from the others?
Discussion
Lines 295 to 301: This first paragraph is too ambitious and does not reflect the study carried out. First of all, the authors should clearly present the results as mean tendencies, that the results were obtained with only 10 pairs of twins and should therefore be regarded as preliminary, and that in some pairs of twins the opposite trend to the general tendency was observed.
Line 397 to 405: In the limitations of the study, I suggest including some possible confounding variables and the possible consequences of not including these variables in the statistical analysis performed. Present future research, taking into account the limitations of your findings.
Lines 406 to 419: Like the first paragraph of the Discussion, the wording of the last paragraph of the Discussion and the paragraph of the Conclusions is too ambitious and should be revised. Future research should be considered in the conclusions, taking into account the limitations of the study carried out.
Abstract: The abstract should be revised after the main text has been revised. I emphasise that the text should reflect that the results obtained with 10 pairs of twins should be regarded as preliminary. I suggest a brief mention of future research.